# Comparative Evaluation of GS-441524, Teriflunomide, Ruxolitinib, Molnupiravir, Ritonavir, and Nirmatrelvir for In Vitro Antiviral Activity against Feline Infectious Peritonitis Virus

**DOI:** 10.3390/vetsci10080513

**Published:** 2023-08-09

**Authors:** Subarna Barua, Bernhard Kaltenboeck, Yen-Chen Juan, Richard Curtis Bird, Chengming Wang

**Affiliations:** Department of Pathobiology, Auburn University College of Veterinary Medicine, Auburn, AL 36830, USA; szb0116@auburn.edu (S.B.); kaltebe@auburn.edu (B.K.); yenchenjuan@gmail.com (Y.-C.J.); birdric@auburn.edu (R.C.B.)

**Keywords:** FIPV, coronavirus, antiviral efficacy, GS441524, teriflunomide, ruxolitinib, molnupiravir, ritonavir, nirmatrelvir

## Abstract

**Simple Summary:**

Developing effective drugs for feline infectious peritonitis (FIP) caused by feline coronavirus (FCoV) is crucial due to its global prevalence and severity in cats. Six antiviral drugs were tested for their cytotoxicity and antiviral efficacies against FCoV in this study. These drugs exhibited minimal to mild cellular toxicity in the cytotoxicity assay. GS441524 and nirmatrelvir exhibited the least detrimental effects on the CRFK cells, with 50% cytotoxic concentration (CC_50_) values of 260.0 µM and 279.1 µM, respectively, while ritonavir showed relatively higher toxicity (CC_50_ = 39.9 µM). In dose–response analysis, GS441524, nirmatrelvir, and molnupiravir demonstrated promising results with selectivity index values of 165.54, 113.67, and 29.27, respectively, against FIPV. Our study suggests that nirmatrelvir and molnupiravir hold potential for FIPV treatment and could serve as alternatives for GS441524.

**Abstract:**

Feline infectious peritonitis (FIP), caused by feline coronavirus (FcoV), is considered one of the most enigmatic diseases in cats. Developing effective drugs for FIP is crucial due to its global prevalence and severity. In this study, six antiviral drugs were tested for their cytotoxicity, cell viability, and antiviral efficacies in Crandell-Reese feline kidney cells. A cytotoxicity assay demonstrated that these drugs were safe to be used with essentially no cytotoxicity with concentrations as high as 250 µM for ruxolitinib; 125 µM for GS441524; 63 µM for teriflunomide, molnupiravir, and nirmatrelvir; and 16 µM for ritonavir. GS441524 and nirmatrelvir exhibited the least detrimental effects on the CRFK cells, with 50% cytotoxic concentration (CC_50_) values of 260.0 µM and 279.1 µM, respectively, while ritonavir showed high toxicity (CC_50_ = 39.9 µM). In the dose–response analysis, GS441524, nirmatrelvir, and molnupiravir demonstrated promising results with selectivity index values of 165.54, 113.67, and 29.27, respectively, against FIPV. Our study suggests that nirmatrelvir and molnupiravir hold potential for FIPV treatment and could serve as alternatives to GS441524. Continued research and development of antiviral drugs are essential to ensure the well-being of companion animals and improve our preparedness for future outbreaks of coronaviruses affecting animals and humans alike.

## 1. Introduction

Feline coronavirus (FCoV), a viral pathogen of cats, is highly contagious and widely prevalent worldwide [1]. This pathogen is usually transmitted through the fecal–oral route, causing mild to subclinical transient gastrointestinal infections in most cats. However, 13% of infected cats remain persistently infected [2]. Out of these persistent cases, approximately 5% experience a mutation of FCoV into feline infectious peritonitis virus (FIPV), resulting in FIP, a fatal disease with a poor prognosis [3]. The progression of FCoV infection into FIP is influenced by various factors, such as virulence and viral burden, as well as the cat’s immune response. Once clinical symptoms such as weight loss, weakness, fever, lethargy, effusion, and ocular lesions manifest, the battle against FIPV has been lost [4]. FIP is considered one of the most enigmatic diseases in cats, and it has been estimated that around 0.3% to 1.4% of feline deaths at veterinary institutions are caused by FIP [5].

There are no FDA-approved treatments, and the only available non-core FIP vaccine (Vanguard^®^ feline FIP intranasal, Zoetis, USA) has limited efficacy and safety. A few unregistered treatment options, such as GS-441524 or GS-441524-like therapy or molnupiravir, are prohibitively expensive as GS-441524 remains unlicensed in USA [2,6,7,8]. Therefore, developing effective drugs for FIP is imperative, considering the global prevalence in cats and the severity of the disease.

GS-441524, the active triphosphate form of Remdesivir, is a 1′-cyano-substituted adenine C-nucleoside ribose analog that inhibits the RNA-dependent RNA polymerase (RdRp), thereby interfering with viral replication (Table 1) [9]. GS-441524 has shown promising results in inhibiting FIPV replication in both experimentally and naturally infected cats, leading to the alleviation of symptoms [5,9,10]. Administration of GS-441524 has been associated with a transient increase in serum globulin levels and led to the resolution of effusions in FIP cases [5,9,11]. In addition, GS441524 co-administered with Remdisivir was found to be an effective and well-tolerated treatment for FIP [6]. Although initially available in Australia and then in the UK in 2021, the novel treatment of FIP with GS-441524 is not yet universally accessible to veterinarians [12].

Teriflunomide, an anti-inflammatory drug used in the treatment of rheumatoid arthritis and other rheumatic conditions (Table 1), acts as a selective inhibitor of dihydroorotate dehydrogenase, a critical mitochondrial enzyme involved in the de novo synthesis of pyrimidines in rapidly proliferating cells. Recent studies showed that continuous teriflunomide treatment in multiple sclerosis patients resulted in milder coronavirus disease 2019 (COVID-19) cases compared to those without treatment [22,23].

Ruxolitinib, a potent inhibitor of JAK1 and JAK2, has demonstrated a good safety profile and is approved for the treatment of myelofibrosis [24] and polycythemia vera [25], both characterized by over-inflammation. In addition, it has shown promising results in rapidly improving severe respiratory conditions associated with COVID-19 when administered in a short-term, high-dose schedule [26].

Molnupiravir (MK-4482 or EIDD-2801), a small molecule with broad-spectrum antiviral activity, has recently received FDA emergency use authorization in the USA for the treatment of symptomatic COVID-19 [27]. This drug inhibited the viral RdRp and was initially developed as an antiviral against different RNA viruses, including influenza [28]. A phase 2a clinical trial of this drug in patients with COVID-19 showed accelerated viral RNA clearance and elimination of infectious viruses [29]. Furthermore, recent reports have shown its efficacy against FIPV [8,17,30]. Cook et al. [8] provided pharmacokinetic analyses of GS-441524, remdesivir, and molnupiravir in cats while treatment of cats with suspected FIP using GS-441524 and molnupiravir was performed by Roy et al. [17].

Ritonavir, an antiretroviral protease inhibitor and CYP3A inhibitor, has been found to enhance the plasma concentration of co-administered drugs [30,31]. Nirmatrelvir, a widely used antiviral drug for COVID-19 [32], acts as a reversible, competitive inhibitor of the FCoV protease [33]. When co-administered with ritonavir within 3 to 5 days of COVID-19 symptom onset, nirmatrelvir has shown a significant reduction in hospitalizations and death by 89% [34].

Despite the absence of approved treatments for FIP in the US thus far, numerous studies have explored commercially available antiviral agents in the hopes of developing effective drugs. In this study, we evaluated six specific antiviral drugs (GS-441524, teriflunomide, ruxolitinib, molnupiravir, ritonavir, and nirmatrelvir) for their in vitro toxicity and antiviral activity against FIPV. By investigating the potential of these drugs, we aim to contribute to the development of novel therapeutic strategies for FIP, addressing the urgent need for effective treatments for this devastating disease.

## 2. Materials and Methods

### 2.1. Cell Line

The Crandell-Reese feline kidney (CRFK) cells utilized in this study were purchased from American Type Culture Collection (ATCC, Manassas, VA, USA), and cultured in Eagle’s Minimum Essential Medium (EMEM) medium supplemented with 10% fetal bovine serum (FBS), 1% antibiotics (penicillin and streptomycin), and 1% non-essential amino acids. The reagents were obtained from ATCC. The CRFK cells were incubated at 37 °C with 5% CO_2_.

### 2.2. Antiviral Drugs

The antiviral drugs GS-441524, teriflunomide, ritonavir, and nirmatrelvir utilized in this study were purchased from MedChemExpress (Princeton, NJ, USA). Ruxolitinib was obtained from InvivoGen (San Diego, CA, USA), while molnupiravir was sourced from Sigma-Aldrich (St. Louis, MO, USA). The antiviral compounds were dissolved or resuspended in dimethyl sulfoxide to prepare stock solutions (5 mM) of all six drugs. The stock solutions of the drugs were stored at −20 °C for ruxolitinib and −80 °C for GS-441524, molnupiravir, nirmatrelvir, ritonavir, and teriflunomide. The drugs were further diluted to working concentrations using EMEM medium for experimental use.

### 2.3. FIPV and Quantification of FIPV via qRT-PCR

The FIPV serotype II strain (WSU-79-1146, GenBank DQ010921) was obtained from ATCC. CRFK cells were utilized to propagate and amplify the FIPV by inoculating the virus in T75 flasks containing EMEM medium supplemented with 10% FBS. After incubation for 72 h at 37 °C, the infected cells displayed extensive cytopathic effects (CPEs) characterized by significant cell detachment.

To ensure maximum virus recovery, the infected flasks underwent three freeze–thaw cycles. The resulting supernatant containing the released virus was then centrifuged at 1500× *g* for 5 min to obtain cell-free viral stocks. These viral stocks were aliquoted and stored at −80 °C for future use. Viral titration was determined using a bioassay (TCID_50_), and the viral load was quantified using real-time reverse transcription polymerase chain reaction (qRT-PCR).

The commercially available High-Pure PCR Template Preparation Kit (Roche Diagnostic, Indianapolis, IN, USA) was used to isolate cell-free total nucleic acid from the collected supernatant following the manufacturer’s instructions. The quantification of FIPV was conducted on a Roche Light Cycler 480 II system (Roche Molecular Biochemicals, Indianapolis, IN, USA) as previously described [35].

### 2.4. The 50% Tissue Culture Infective Dose (TCID_50_) of FIPV Stock

The titration of FIPV was performed using a bioassay method known as the TCID_50_ (median tissue culture infectious dose) assay. CRFK cells were cultivated in a 96-well tissue culture plate (Nunc, ThermoFisher Scientific) until they reached 80–90% confluency. Subsequently, 100 µL of 10-fold serially diluted FIPV stock was added to each well, with six replicates for each dilution. Negative controls consisted of CRFK cells without FIPV infection, while the undiluted FIPV stock was a positive control. Cytopathic effects (CPEs) characterized by rounding of cells, were observed in wells inoculated with virus dilutions as well as in the positive control wells under an inverted phase-contrast microscope. The TCID_50_ endpoint values were then calculated using the CPE scores according to the method of Reed and Muench [36].

### 2.5. Cytotoxicity of the Antiviral Drugs

CellTox Green Cytotoxicity Assay (Promega, Madison, WI, USA) was used to determine the cytotoxicity of six antiviral drugs. CRFK cells at a density of 5 × 10^4^ cells/well were seeded in 96-well plates and incubated at 37 °C and were treated in four-well replicates with 1000, 500, 250, 125, 63, 31, 16, 8, and 4 µM concentrations of the drug at 90% confluency. After 48 h of incubation, the DNA binding dye from the kit was applied to all wells and incubated shielded from light at 37 °C for 15 min. Cytotoxicity of the drugs was measured using a plate reader, SpectraMax iD3 (Molecular Devices, San Jose, CA, USA), with fluorescence intensity at 495/519 nm (λex/λem). The fluorescence of CRFK cells was compared to the untreated CRFK cells as the negative control and lysing-reagent-treated cells as the positive control. The fluorescence reading is proportional to cell death due to the selective binding of the DNA of apoptotic/necrotic cells to the dye. The mean fluorescence value of all four replicates for each drug concentration was interpolated as percent cytotoxicity (%) ranging from 0 to 100%, where untreated cells were considered as 0% (baseline cytotoxicity) and cells treated with the positive control reagent as 100%.

### 2.6. Cell Viability Assay of the Antiviral Drugs

The cell viability of CRFK cells was measured using a Cell Proliferation Kit I (MTT) (Roche Applied Science, Indianapolis, IN, USA), following the manufacturer’s instructions. Cells were seeded at 5 × 10^4^ cells/well into 96-well plates and incubated at 37 °C and were treated in four-well replicates with 1000, 500, 250, 125, 63, 31, 16, 8, and 4 µM concentrations of the drug of interest at 90% confluency. After incubation for 48 h, 10 μL of MTT solution (0.5 mg/mL in phosphate-buffered saline) was added to each well. The plate was incubated for 4 h at 37 °C to allow the formation of formazan crystals. Following incubation, 1% sodium dodecyl sulfate (100 μL) solution was added to dissolve the crystals. Then, the spectrophotometrical absorbance of the sample was measured at 575 nm wavelength using a microplate reader, SpectraMax iD3 (Molecular Devices, CA, USA), as described [37].

### 2.7. Antiviral Efficacies of the Drugs

CRFK cells were seeded in 96-well plates at a density of 5 × 10^4^ cells per well and incubated at 37 °C. When the cells reached approximately 90% confluency, the culture media was discarded, and the cells were infected with FIPV at 2.5 × 10^4^ TCID_50_/mL. The culture plate was then incubated for one hour with occasional gentle agitation every 15 min to allow virus attachment and entry. After the incubation period, the medium containing the unbound virus was removed, and fresh EMEM medium supplemented with 2% FBS, containing various dilutions of the antiviral drugs, was added to the wells. Wells containing CRFK cells infected with FIPV but without any drug treatment served as the positive control, while wells without virus infection or drug treatment acted as the negative control. The plate was subsequently incubated at 37 °C in a 5% CO_2_ atmosphere for the desired duration specified by the experiment. Following incubation, the culture supernatant was collected and stored for further virus quantification and analysis via qRT-PCR.

### 2.8. Statistical Analysis

All statistical analyses were performed using Statistica 7.0 software package (StatSoft, Inc., Tulsa, OK, USA). Shapiro–Wilk’s W test confirmed the normal distribution of data, and Levene’s test confirmed the homogeneity of variances. Data were analyzed using mean plots with 95% confidence intervals (CI) in one-way or factorial analysis of variance. Comparisons of means under the assumption of no a priori hypothesis were performed using a two-tailed Tukey honest significant difference (HSD) test. Differences at *p* values of 0.05 in the Tukey HSD test were considered significant.

## 3. Results

### 3.1. Effect of Initial Inocula and Incubation Times on Anti-FIPV Efficacy of GS441524

Using an endpoint dilution assay, the virus titer was evaluated as 2.5 × 10^4^ TCID_50_/mL in FIPV stock. The initial virus inoculum and the length of incubation time can influence the in vitro antiviral efficacy. Here, we chose GS441524 as an example to examine the impact of these factors on the anti-FIPV efficacy in CRFK cells (Table 2).

CRFK cells with 80–90% confluency were infected with three FIPV inocula: 2.5 × 10^4^, 2.5 × 10^3^, and 2.5 × 10^2^ TCID_50_/mL. The cells were then incubated in the presence or absence of GS441524 molecules (25 µM), and were evaluated at 48 h and 72 h (Table 2). Although variations in viral copy numbers were observed with different inocula and incubation times, the overall trend and conclusion regarding GS441524’s antiviral efficacy remained consistent, regardless of the initial inoculum and incubation time. This result demonstrated that the GS441524 molecule is very effective against FIPV irrespective of the initial viral dose used in this experiment inoculated for at least 72 h. Therefore, 48 h of incubation with the highest inoculum of FIPV (2.5 × 10^4^ TCID_50_/mL) was used in the remaining assays.

### 3.2. Effect of Different Addition and Removal Times of GS441524 on the Anti-FIPV Efficacy

In the context of antiviral drugs used for treating viral infections, drugs can be administered at different stages of infection, with varying doses and frequencies. To mimic real-life scenarios, we designed an experiment involving the addition and removal of GS441524 at five different time-points (Figure 1). The five groups in this study were as follows: no drug: no drug was applied; +1 hpi: the drug was applied 1 h post inoculation; −2 hpi: the drug was applied 2 h before inoculation and was removed before virus inoculation; −2 & +1 hpi: the drug was applied 2 h before infection and was removed before virus inoculation and applied again at 1 hpi; 0 hpi: the drug was applied at the time of inoculation and remained in the wells.

Confluent monolayers of CRFK cells were inoculated with FIPV (2.5 × 10^4^ TCID_50_/mL) for 1 h. Compared to the control wells (no drug), the inhibition of FIPV replication in the presence of GS441524 (25 µM) in CRFK cells did not differ up to 72 hpi with different initial drug application time-points, except for the −2 hpi group (Table 3). In the latter group, the FIPV RNA copies were significantly higher at 72 h (275-fold) than at 24 (0.8-fold) and 48 h (11.8-fold). The peak viral load of the −2 hpi wells at 72 h of incubation is similar to the no-drug group at 48 h, indicating that the initial antiviral activity observed results from limited initial inhibition by GS441524.

The results indicated that antiviral efficacy did not differ significantly if GS441524 was applied during or after FIPV infection, but there was a significant drop in efficacy when the drug was applied 2 h before inoculation and was removed before virus inoculation was removed. The drug was applied at +1 hpi in the following viral efficacy experiments.

### 3.3. Cytotoxicity and Cell Viability of Antiviral Drugs

Except for ritonavir, all antiviral drugs showed no cytotoxicity up to 63 µM (Figure 2). Ritonavir exhibited toxicity to cells at a concentration as low as 31 µM. Conversely, ruxolitinib demonstrated undetectable cytotoxicity even at a high concentration of 250 µM. GS441524 was found to be non-toxic to cells up to a concentration of 125 µM.

The analysis of MTT staining in CRFK cells treated with various doses of the different drugs revealed a dose-dependent decrease in cell viability. These findings indicated that the six drugs could be used safely with minimal cytotoxicity at concentrations as high as 125 µM for GS441524, 62.5 µM for teriflunomide, 250 µM for ruxolitinib, 63 µM for molnupiravir, 16 µM for ritonavir, and 63 µM for nirmatrelvir.

Overall, the cell viability data inversely corresponded with the cytotoxicity of the drugs except at high drug concentrations (Appendix A).

### 3.4. Antiviral Efficacies of Six Antiviral Drugs

A dose–response analysis was performed on CRFK cells infected with FIPV (2.5 × 10^4^ TCID_50_/mL) and treated with 10-fold dilutions of the drugs to assess their antiviral efficacy ranges (Appendix A). The viral load of harvested FIPV was measured using qRT-PCR at 48 h post infection.

All six drugs exhibited anti-FIPV efficacy starting from a concentration as low as 5 µM compared to the no-drug group (Figure 3). GS441524 demonstrated effectiveness against FIPV at a concentration as low as 0.5 µM, resulting in 85% inhibition of FIPV replication and no CPEs observed in cells. Nirmatrelvir also displayed promising anti-FIPV efficacy, with 80% inhibition of FIPV replication at a concentration as low as 5 µM and no CPEs in cells. Teriflunomide and molnupiravir exhibited dose-dependent antiviral activities. However, ritonavir showed a sudden decline in FIPV antiviral efficacy when the concentration decreased from 50 µM to 5 µM, accompanied by an increase in the FIPV copy number by up to two logarithmic values (Appendix A).

To further assess the antiviral efficacies of the six drugs, a serial two-fold dilution was performed to determine the concentration range at which they remained effective against FIPV in CRFK cells (Figure 4). Based on the results obtained from this experiment, GS441524 demonstrated antiviral efficacy starting from a concentration as low as 0.98 µM, resulting in a 22.5% reduction in FIPV copy number. A significant inhibition of FIPV (98.3%) was observed at a concentration of 7.8 µM, with no CPE observed in cells (Appendix A). Following GS441524, nirmatrelvir exhibited 98.3% inhibition of FIPV replication at the same concentration without inducing cell CPEs.

### 3.5. Selectivity Index (SI) of Six Drugs

The 50% cytotoxic concentration (CC_50_), 50% effective concentration (EC_50_), and selectivity indices (SI) were calculated for the six drugs (Figure 5). GS441524 and nirmatrelvir exhibited the least detrimental effects on CRFK cells, with CC_50_ values of 260.0 µM and 279.1 µM, respectively. On the other hand, ritonavir demonstrated high cytotoxicity with a CC_50_ of 39.9 µM in this in vitro study.

When the selectivity index (SI) was calculated by dividing the CC_50_ by the EC_50_, GS441524 displayed the highest SI value (165.5) among the drugs tested against FIPV. Nirmatrelvir also exhibited promising anti-FIPV selectivity with an SI value of 113.7. Molnupiravir showed a relatively safe profile and selectivity against FIPV with an SI of 29.3 and an EC_50_ of 8.0 µM. However, the remaining three drugs were either cytotoxic (ritonavir) or less effective against FIPV (teriflunomide, ruxolitinib), resulting in lower selectivity towards FIPV (SI values of 2.3, 0.4, and 7.8, respectively). In summary, among the six drugs, GS441524 and nirmatrelvir demonstrated promising results in terms of safety and higher efficacy against FIPV.

## 4. Discussion

The development of antiviral drugs targeting FCoV replication is of great importance, particularly in the absence of a safe and approved vaccine for FIPV infection. In the case of antiviral drug development, it is crucial to target viral enzymes, essential for replication, while minimizing adverse effects on uninfected cells. In this study, six drugs were selected and evaluated for their anti-FIPV activity, and these drugs were selected based on their antiviral efficacy profiles and available data on their effectiveness against FIPV and SARS-CoV-2 (Table 1).

The cell viability data should correspond inversely with the cytotoxicity of the drugs. However, there was a discrepancy between the drug cytotoxicity and cell viability data at 277 high concentrations. The possibility was that an overall reduction in the cell number occurred within the drug-treated wells due to the toxicity of high drug concentrations, which resulted in the degradation and loss of nucleic acids and made them unavailable for fluorescence binding and detection in the CellTox assay.

Among the six drugs evaluated in the study, the qRT-PCR results indicate that GS441524 exhibited the highest effectiveness against FIPV while maintaining safety, as reflected by its high selectivity index (SI) of 165.54. Following GS441524, nirmatrelvir and molnupiravir also demonstrated significant anti-FIPV activity with SI values of 113.67 and 29.27, respectively. However, teriflunomide, ruxolitinib, and ritonavir exhibited relatively low selectivity for FIPV with SI values of 2.7, 7.8, and 2.3, respectively. These drugs showed less-favorable profiles in terms of both efficacy and safety compared to GS441524, nirmatrelvir, and molnupiravir. Interestingly, ritonavir induced CPEs at both 50 µM and 5 µM concentrations, with cytotoxic effects observed at concentrations as low as 31 µM in the cytotoxicity assay. Consequently, it can be speculated that the lower FIPV copy number of ritonavir at 50 µM may be attributed more to cytotoxicity than to drug efficacy. A similar observation applies to ruxolitinib, where a change in drug concentration from 500 µM to 50 µM resulted in a two-log value increase in FIPV copy number, accompanied by cytotoxicity observed at a concentration as low as 250 µM.

The findings of this study support the previous research on GS441524, which has demonstrated its safety and efficacy as a treatment for FIP [5,9]. The calculated EC_50_ value of 1.57 µM and CC_50_ value of 260.02 µM further validate the drug’s potency against FIPV while maintaining a favorable safety profile. GS441524 was found to be safe and effective in other in vitro and in vivo studies. Pederson et al. showed GS-441524 to be a safe and effective treatment for FIP with the optimum dosage to be 4.0 mg/kg dose [5].

Nirmatrelvir, an FDA-approved protease inhibitor originally developed for SARS-CoV-2, has shown its efficacy by targeting the viral protease (Mpro) and interfering with viral replication [38]. In this study, nirmatrelvir demonstrated a favorable safety profile with a CC_50_ value of 279.1 µM, indicating low toxicity to CRFK cells. Its antiviral efficacy against FIPV was also evident, as indicated by an EC_50_ value of 2.5 µM. The calculated selectivity index (SI) of 113.67 further highlights its specificity and potential therapeutic value for FIPV treatment. These observations are consistent with previous studies that have reported the inhibitory effect of nirmatrelvir on FIPV. Notably, when investigating mutation sites associated with CoV resistance to Mpro inhibitors, Jiao, Yan et al. also observed a significant inhibitory effect (EC50 2.52 µM) of nirmatrelvir on FIPV [39]. Drawing from the experience with the closely related GC376, nirmatrelvir holds promise as an oral treatment option for FIP in the future [10,40]. The encouraging results obtained with nirmatrelvir in this study suggest its potential as a viable therapeutic option for FIPV infections, warranting further investigation and potential clinical application in the management of FIP.

The findings from this study suggest that molnupiravir could be a potential alternative to GS441524 for treating FIPV. Molnupiravir has gained attention as a potential therapeutic option, although it is available only in the unapproved market. In this study, molnupiravir demonstrated effectiveness against FIPV with an EC_50_ value of 8.04 µM. This indicates its ability to inhibit FIPV replication in CRFK cells. Furthermore, the drug exhibited a non-toxic profile with a CC_50_ value of 235.35 µM, suggesting a relatively safe range of concentrations for use in vitro. However, it is important to note that molnupiravir’s effective dosage recommendation for FIPV is based on presumptions drawn from published information on its use in COVID-19 treatment [41,42]. Further experience and research specifically focused on FIPV are needed to fully understand its efficacy and safety profile in the context of FIP.

One concern associated with molnupiravir is the potential mutagenic properties of its active metabolite, N4-deoxycytidine. Although molnupiravir has shown non-toxicity to CRFK cells, the mutagenic nature of its metabolite raises questions about the possibility of side effects during prolonged treatment, as the treatment duration for FIP is longer compared to COVID-19 [43]. This possibility highlights the need for additional investigation to evaluate the long-term safety and potential side effects of molnupiravir in the context of FIP treatment. While molnupiravir showed promise as a potential treatment option for FIPV, further research is necessary to fully assess its efficacy, safety, and potential risks before it can be considered a recommended and approved treatment for FIPV infections.

While paxlovid, a novel drug containing nirmatrelvir and ritonavir, has shown efficacy against SARS-CoV-2 in humans [44,45], further research is needed to determine its effectiveness and safety, specifically for FIPV in cats. Conducting studies to evaluate the efficacy and potential side effects of nirmatrelvir and paxlovid in the context of FIP treatment would provide valuable insights into their potential as oral therapies for FIPV.

Taken together, it is important to continue exploring and developing antiviral drugs for veterinary coronaviruses like FIPV to ensure the well-being of companion animals and to enhance our preparedness for potential future outbreaks of coronaviruses affecting animals and humans alike.

## 5. Conclusions

The findings from this FIPV antiviral efficacy study are promising, indicating that GS441524, nirmatrelvir, and molnupiravir are safe and effective in inhibiting FIPV replication. These results suggest that nirmatrelvir and molnupiravir could provide new possibilities for treating FIPV in addition to GS441524. Moreover, they may serve as potential alternatives for treating FIPV strains that have developed resistance to GS441524 in cats. While the in vitro antiviral efficacies of these drugs are encouraging, further research is needed to assess their treatment efficacies in vivo and investigate any potential side effects in FIP therapy. Conducting additional studies, such as animal trials or clinical trials, will provide a more comprehensive understanding of the effectiveness and safety profiles of these drugs in the treatment of FIP in cats. These promising results warrant continued exploration of GS441524, nirmatrelvir, molnupiravir, and other potential antiviral drugs for the development of effective treatments against FIPV. Continued research in this area will contribute to advancing FIP therapy and improving the outcomes for cats affected by this devastating disease.

## Figures and Tables

**Figure 1 vetsci-10-00513-f001:**
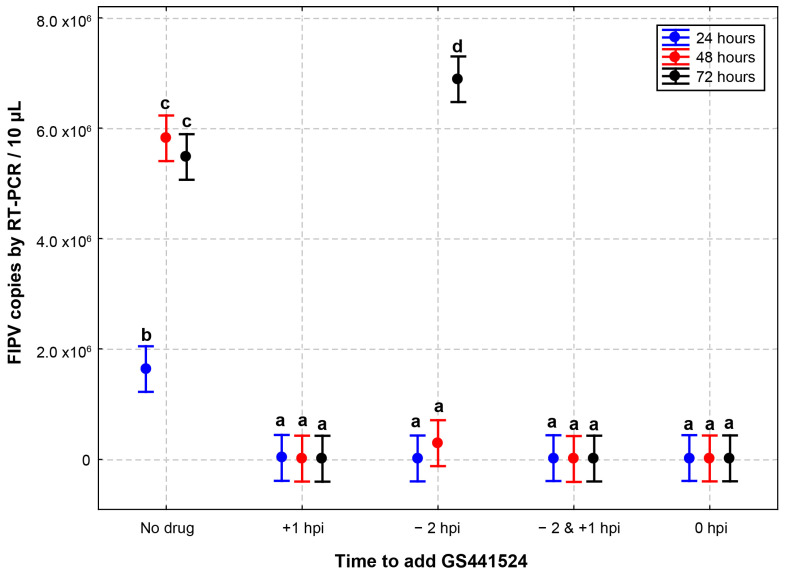
**Effect of different addition and removal times of GS441524 on its anti-FIPV efficacy.** CRFK cells infected with FIPV (2.5 × 10^4^ TCID_50_/mL) received GS441524 (25 µM) with different times of addition and removal. No drug: no drug was applied; +1 hpi: the drug was applied 1 h post inoculation; −2 hpi: the drug was applied 2 h before inoculation and was removed before virus inoculation; −2 & +1 hpi: the drug was applied 2 h before infection and was removed before virus inoculation and applied again at 1 hpi; 0 hpi: the drug was applied at the time of inoculation and remained in the wells. The inhibition of FIPV replication in the presence of GS441524 in CRFK cells did not differ with different initial drug application time-points for as long as 72 h, except for the −2 hpi well. In the −2 hpi wells, the FIPV RNA copies were significantly higher at 72 h than at 24 and 48 h. These data are presented as an average ± 95% CI. Different letters (a, b, c, and d) indicate significant differences in FIPV RNA copies. *P* value below 0.05 is considered statistically significant.

**Figure 2 vetsci-10-00513-f002:**
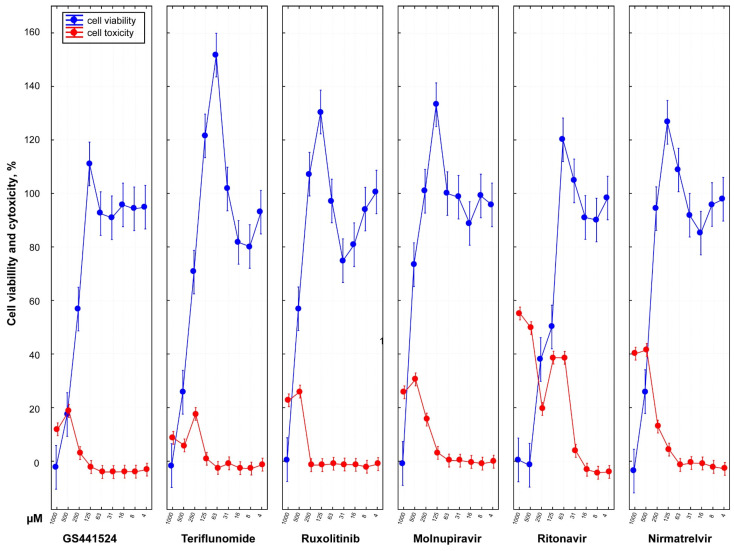
**Dose-dependent cell viability and cytotoxicity profile of six antiviral drugs.** In vitro analysis of cytotoxicity of different drugs was performed using commercially available CellToxTM Green assay for CRFK cells treated with serially diluted two-fold medicines (1000, 500, 250,125, 63, 31, 16, 8, 4 µM) with four well replicates. At 48 hpi, drug cytotoxicity was calculated with a fluorescence intensity of 485–500 nm_Ex_/520–530 nm_Em_. In the graph, the y-axis presents the percent cytotoxicity determined by normalizing sample cytotoxicity to the positive toxicity control (100% cytotoxicity) and untreated CRFK cells (0% baseline cytotoxicity). The plotted data (red line) represent the average ± 95% CI of four replicates for each treated diluted drug concentration. All six antiviral drugs except ritonavir demonstrated no cytotoxicity up to 63 µM. Ritonavir was found to be highly toxic to cells with CC_50_ of 39.85 µM. In contrast, ruxolitinib showed a safer cytotoxicity profile undetectable at a high 250 µM concentration with CC_50_ of 338.05 µM compared to the other drugs. The value of CC_50_ of GS441524 had been determined as 260.02 µM. Cell viability assay was also performed using Cell Proliferation Kit I (MTT assay) for CRFK cells. Cell viability data inversely correspond with the drug cytotoxicity except at the highest drug concentration.

**Figure 3 vetsci-10-00513-f003:**
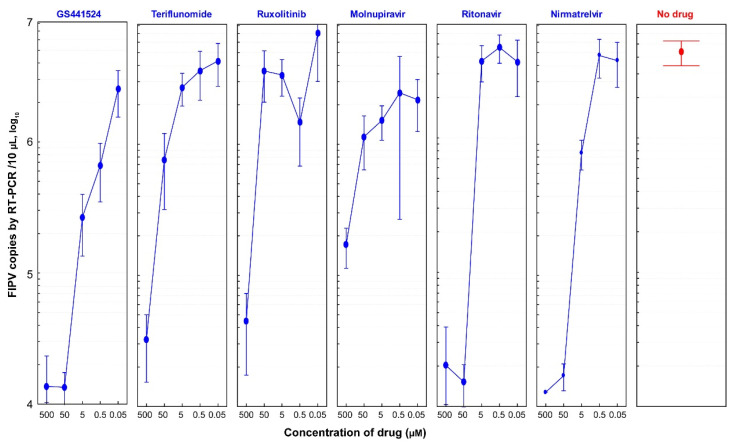
**Dose-dependent anti-FIPV efficacy of antiviral drugs (1:10 dilutions).** CRFK cells were infected with FIPV (2.5 × 10^4^ TCID_50_/mL) and treated with 10-fold dilutions of six drugs (GS441524, teriflunomide, ruxolitinib, molnupiravir, ritonavir, and nirmatrelvir). The viral load of FIPV was quantified by qRT-PCR at 48 hpi. Data represent an average ± 95% CI of four replicates in the y-axis for each indicated concentration of six drugs. Compared to the no-drug group, all tested drugs showed anti-FIPV efficacy at a concentration as low as 5 µM. GS441524 molecule showed the most promising anti-FIPV efficacy at a concentration as low as 0.5 µM. Teriflunomide, molnupiravir, and nirmatrelvir harbored dose-dependent antiviral activities. Interestingly, ritonavir showed a sudden drop in FIPV antiviral efficacy when the concentration changed from 50 µM to 5 µM.

**Figure 4 vetsci-10-00513-f004:**
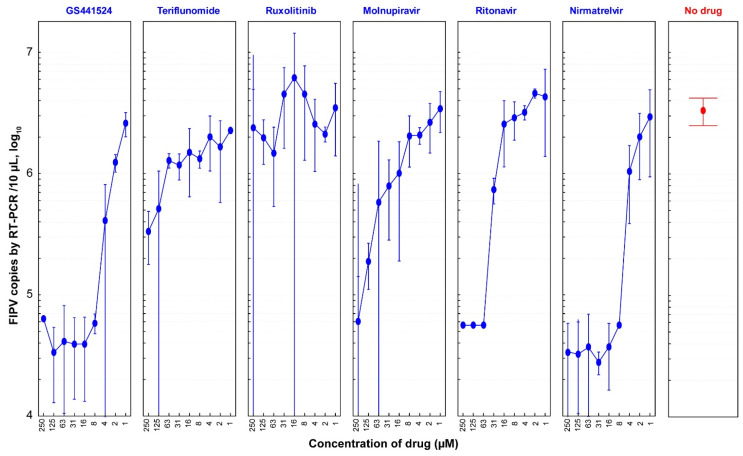
**Dose-dependent anti-FIPV efficacy of antiviral drugs (1:2 dilutions).** CRFK cells were infected with FIPV (2.5 × 10^4^ TCID_50_/mL) and treated with two-fold dilutions of six drugs (GS441524, teriflunomide, ruxolitinib, molnupiravir, ritonavir, and nirmatrelvir). At 48 hpi, the viral load of FIPV was quantified via qRT-PCR. In this graph, FIPV copies in the y-axis represent the average ± 95% confidence interval (CI) of four replicates of samples for each diluted drug concentration. GS441524 and nirmatrelvir showed significant anti-FIPV efficacy with EC_50_ values of 1.57 µM and 2.46 µM, respectively.

**Figure 5 vetsci-10-00513-f005:**
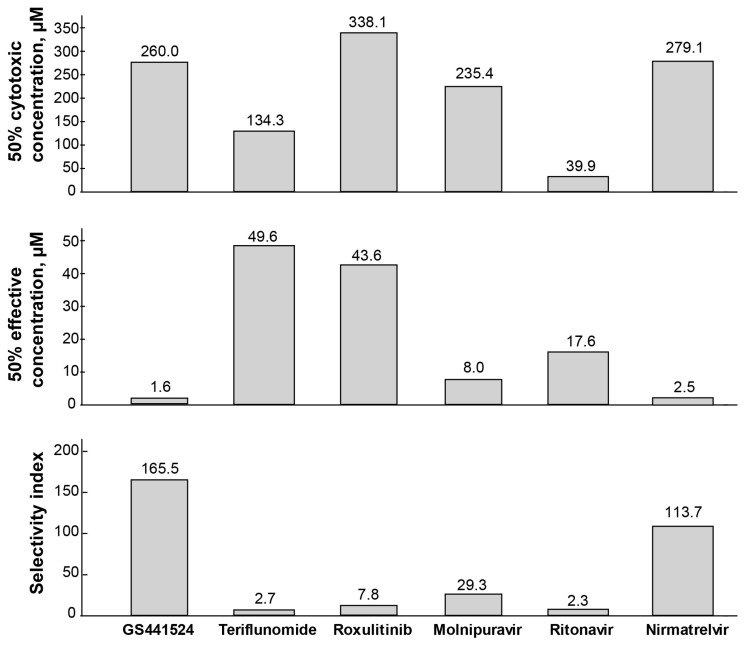
**The CC_50_, EC_50_, and SI for six drugs against FIPV.** The half-maximal cytotoxic concentration (CC_50_) values are from four measurements of diluted drugs using MTT assay, in CRFK cells treated with drugs for 48 h. The half-maximal effective concentration (EC_50_) values are from six measurements of diluted drugs against FIPV replication in CRFK cells for 48 h. Based on the SI value (mean CC_50_)/(mean EC_50_), GS441524 was found highly selective (SI 165.5) against FIPV among the drugs tested and showed high efficacy (EC_50_ 1.6 µM) against FIPV with a less deleterious effect (CC_50_ 260.0 µM) on the cells. Nirmatrelvir also showed promising efficacy (EC_50_ 2.5 µM) and selectivity (SI 113.7) against FIPV. Ritonavir showed the highest toxicity level in the cells (CC_50_ 39.9).

**Table 1 vetsci-10-00513-t001:** List of drugs with antiviral activity against animal and human coronaviruses (CoVs) used in this study.

Drug Name	Drug Category	Viral Target	Investigated CoVs *	References
GS441524	Adenosine nucleotide analog	Inhibition of RNA systhesis	FIPV, SARS-CoV-2, MERS-CoV	[5,6,9,13,14]
Nirmatrelvir	3C-like Protease inhibitor	Inhibition of RNA systhesis	FIPV, SARS-CoV-2	[15]
Molnupiravir	Isopropyl ester cytidine analog	Inhibition of RNA systhesis	FIPV, SARS-CoV-2, MERS-CoV	[8,16,17]
Ruxolitinib	Kinase inhibitor	Inhibition of viral entry	SARS-CoV-2	[18]
Ritonavir	Antiretroviral protease inhibitor	Cleavage of viral polyproteins	FIPV, SARS-CoV-2	[19,20]
Teriflunomide	Dihydroorotate dehydrogenase inhibitor	Inhibition of RNA systhesis	SARS-CoV-2	[21]

* SARS-CoV-2: severe acute respiratory syndrome coronavirus 2; MERS-CoV: Middle East respiratory syndrome coronavirus.

**Table 2 vetsci-10-00513-t002:** Effect of GS441524 molecule with different inocula and incubation times on FIPV replication.

FIPV Inoculum	48 h			72 h		
	No Drug	GS441524	Fold Reduced	No Drug	GS441524	Fold Reduced
2.50 × 10^4^ *	5.91 × 10^6^	2.76 × 10^3^	2.14 × 10^3^	4.67 × 10^6^	3.24 × 10^3^	1.44 × 10^3^
2.50 × 10^3^	1.17 × 10^7^	3.48 × 10^2^	3.36 × 10^4^	5.18 × 10^6^	2.09 × 10^2^	2.47 × 10^4^
2.50 × 10^2^	1.23 × 10^7^	4.60 × 10^1^	2.67 × 10^5^	9.05 × 10^6^	2.36 × 10^1^	3.83 × 10^5^

* The unit of viral copies in this table and this study was per ml while 0.1 mL of the inoculum was used.

**Table 3 vetsci-10-00513-t003:** Effect of different addition times of GS441524 on the replication of FIPV.

Treatments *	Increase in Viral Load					
	24 h	Fold Increase **	48 h	Fold Increase	72 h	Fold Increase
−2 hpi ***	1.97 × 10^4^	0.79	2.94 × 10^5^	11	6.89 × 10^6^	275
−2 & +1 hpi	2.47 × 10^4^	0.99	1.03 × 10^4^	0.41	1.62 × 10^4^	0.65
0 hpi	2.58 × 10^4^	1.03	1.97 × 10^4^	0.79	2.04 × 10^4^	0.82
+1 hpi	2.84 × 10^4^	1.14	1.52 × 10^4^	0.61	1.39 × 10^4^	0.56
No drug	1.63 × 10^6^	65	5.82 × 10^6^	232	5.48 × 10^6^	219

* hpi = hours post infection; ** fold increase was calculated based on the initial FIPV inoculum (2.5 × 10^4^ TCID_50_/mL) used in this experiment for all treatments. *** No drug: no drug was applied; +1 hpi: the drug was applied 1 h post inoculation; −2 hpi: the drug was applied 2 h before inoculation and was removed before virus inoculation; −2 & +1 hpi: the drug was applied 2 h before infection and was removed before virus inoculation and applied again at 1 hpi; 0 hpi: the drug was applied at the time of inoculation and remained in the wells.

## Data Availability

The data supporting the conclusions of this article are included within the article. Raw data are available from the corresponding author upon reasonable request.

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
