# Peer review of "Comparative Evaluation of GS-441524, Teriflunomide, Ruxolitinib, Molnupiravir, Ritonavir, and Nirmatrelvir for In Vitro Antiviral Activity against Feline Infectious Peritonitis Virus"

_vetsci, 2023, doi:10.3390/vetsci10080513_

Round 1
Reviewer 1 Report
Overview of this study: This is a interesting manuscript evaluating six antiviral drugs for the use in treating feline infectious peritonitis. In this study, feline kidney cell lines were infected with FIPV and treated with 6 different antiviral drugs. FIPV viral replication and cellular toxicity were assessed. The study shows that all the antiviral drugs decreased FIPV replication and 5/6 exhibited minimal to mild cellular toxicity. The data is well presented. Statistical methods are not described or reported which is a major flaw that needs to be addressed.
Specific comments:
Lines 39-40: Please cite the APA pet owners survey
Lines 29-43: Unclear the significance of listing these studies for this manuscript
Lines 48-49: Could examples of these treatments be added to this paragraph?
Line 58: “Effusion” should be included in the list of clinical symptoms listed in lines 37-38
Line 105: What is “cell culture maintenance media”?
Lines 109-110: Where were the cell culture media components obtained from?
Line 175: Please list the range of virus inoculum used.
Lines 179-180: Please list the concentration of the drugs used. “Dilations based on the experimental requirements” is confusing to the reader.
Line 185: Viral quantification and analysis via what methods?
Line 197: What specifically was evaluated in these cells?
Figure 1, line 226: Please state the p value. What method was used to determine statistical differences?
Lines 276-280: Are pictures of the CPE described available? This reviewer would like these included as a supplemental figure to address this discrepancy between the drug cytotoxicity and cell viability that is described.
Table 4: This table is very difficult to read. Please revise.
Table 5: This table is very difficult to read. Please revise.
Line 372: Table 1 only lists investigated CoVs as FIPV, SARS-CoV-2, and MERS-CoV. FCoV has not been evaluated based on the literature cited in Table 1 and “FCoV” should be changed to FIPV.
General:
Methods: methods statement for statistics is needed
This reviewer would like to ask the authors to address/speculate in the discussion about the use of these drugs in vivo. Specific questions to consider include:
1. What type of tissue/blood concentration of these drugs can be achieved in animals or humans? Are these tissue concentrations relevant to the concentrations tested in vitro in this study?
2. How are these drugs metabolized and secreted? How might that affect the in vivo use and toxicity? For example, many older cats have chronic renal disease or sick anorexic cats can develop hepatic lipidosis.
Overall the manuscript was well written, there are minor grammar and flow editing is needed.
Specific comments:
Lines 48-49: As currently written, this paragraph followed up a list of the drugs feels choppy.
Line 108: Extra “d” after purchased.
Lines 284-287: This sentence is confusing, please revise.
Lines 381, 385, 401: These three paragraphs all start with the same phrase. Please revise.
Author Response
Please see the attached file, thanks!

Reviewer 2 Report
The authors described that GS441524, nirmatrelvir, and molnupiravir were safe and effective in inhibiting FIPV replication, suggesting that nirmatrelvir and molnupiravir could provide new possibilities for treating FIPV in addition to GS441524. However, this study has largely lost novelty after the recent publications by Cook et al. (PMID: 36366527) and Roy et al.(PMID: 36297266).
Specific points:
1. Line 53. RNA-dependent RNA polymerase (RdRp).
2. Table 1. Ref [5] should be included. SARS-CoV-2 and MERS-CoV should be spelled out.
3. Line 68. COVID-19 should be spelled out.
4. Line 107. Delete were.
5. Line 109. EMEM ans FBS should be spelled out.
6. Line 113. Delete (ATCC, Manassas, VA, USA).
7. Lines 155-171. Replace with MTT assay.
8. Tables 4 and 5. It is unclear what the authors mean by these Tables.
9. Lines 386-388. Nirmatrelvir does never interfere with virus entry into host cells.
As there are many errors, the authors should carefully check the text.
Author Response
Please see the attached file, thanks!

Reviewer 3 Report
Line 12: most enigmatic diseases in cats
Line 32: transient gastrointestinal infections in most cats.
Line 39- 43: "According to the American Pet Products Association (APPA) National Pet Owners Survey conducted in 2023-2024, there are approximately 46.5 million pet cats in the United States. In addition, based on estimates from the American Society for the Prevention of Cruelty to Animals (ASPCA), there are tens of millions of feral cats in the country."
This two sentences do not seem to be too relevant to FIP, I would suggest to delete them.
Line 43: most enigmatic diseases in cats
Line 46- 48: This sentence is confusing. "Lack a safe and effective vaccine, except..." implicates that this vaccine is safe and effective. But the end of the sentence said it has limited use. Please re-phrase this sentence.
Table 1 and introduction: Remdesivir has recently gained increasing attention in the treatment of FIP, is an effective and well tolerated treatment when used with GS. Remdesivir should be discussed (Coggins & Thompson 2023).
Line 284- 287 should be in the discussion part to explain these changes, rather than in results section.
Line 311: teriflunomide (capital)
Line 316- 320 should be in the discussion part to explain these changes, not in results
Good, minor editing required (see comments).
Author Response
Please see the attached file, thanks!

Round 2
Reviewer 2 Report
Although the manuscript has been improved, the findings still represent a limited advancement in this area of research.
Specific points:
1. Lines 52-53. This sentence should be revised.
2. Table 1. The expression of replication inhibition is not appropriate. RNA synthesis, cleavage of viral poly protein, or something should be used.
3. Line 182 and others. Did the authors used 2.5x104 or 103 of virus as a inoculate? They added 0.1ml to the well.
4. Fig. 3 should be deleted, since this experiment is a preliminary one of the experiments in Fig. 4.
5. Fig.4. The title should be revised.
6. Fig. 5. Remove CC50, EC50, and SI from each panel, respectively.
7. References. Please check 6, 7, 12, 28, and so on.
There are a small number of typographical errors -- e. g. line 49, It.
Author Response
Please see the attached file, thanks!
